# An Advanced Edge-Detection Method for Noncontact Structural Displacement Monitoring

**DOI:** 10.3390/s20174941

**Published:** 2020-09-01

**Authors:** Xin Bai, Mijia Yang, Beena Ajmera

**Affiliations:** Department of Civil and Environmental Engineering, North Dakota State University, Fargo, ND 58108, USA; xin.bai@ndsu.edu (X.B.); beena.ajmera@ndsu.edu (B.A.)

**Keywords:** vision sensor system, edge detection, Zernike matrix, subpixel resolution, mode shape

## Abstract

A non-contact vision sensor system for monitoring structural displacements with advanced Zernike subpixel edge detection technique is suggested in this paper. Edge detection can detect features of objects effectively without using templates. Subpixel techniques provide more accurate and cost-effective results when compared to integer pixel methods. Built on these two techniques, a new version sensor method was developed to detect the vibrations of structures in this study. Satisfactory agreements were found between the displacements measured by the vision sensor system and those recorded by the Multipurpose Testing System (MTS). A field test was then carried out on a street sign using the proposed vision system. Satisfactory results were obtained using the new version of the sensor system at many points simultaneously without any manually marked targets. Moreover, the system was able to provide natural frequencies and mode shapes of the target instantaneously, which could be used to accurately locate damage.

## 1. Introduction

Bridges are exposed to many external loads, such as from traffic, wind, flooding, and earthquakes. Monitoring bridges is becoming increasingly important due to safety concerns. There are some direct measurement methods such as the use of linear variable differential transformer (LVDT) and laser-based displacement sensors. However, many LVDTs are required to measure the displacement across a bridge, which can be both costly and time-consuming. Another method for obtaining displacements is through a non-contact method, such as using GPS, but the precision of GPS displacement measurements is not satisfactory. Nassif et al. [1] compared the displacement results of a girder using a non-contact laser Doppler vibrometer (LDV) system with those from a linear variable differential transducer (LVDT) system (deflection) and geophone sensors (velocity), both attached to the girder. Similar accuracy was reached. Gentile and Bernardini [2] proposed a new radar system, called IBIS-S, which was used to measure the static or dynamic displacements at several points on a structure. Kohut et al. [3] compared the structure’s static displacements under a load using the digital image correlation method with those from the radar method. The main advantage of the digital image correlation method is that it is cheaper and easier to obtain the displacements of multiple points simultaneously. Fukuda et al. [4] proposed a vision-based displacement system to detect the displacement of large-scale structures. Their system requires a low-cost digital camcorder, a notebook computer, and a target panel with predesigned marks. Ribeiro et al. [5] developed a non-contact dynamic displacement sensor system based on video technology to detect the displacements in both laboratory and field settings. Both works indicated similar or higher accuracy in displacement measurements compared to traditional methods such as LVDT measurements.

To adopt a computer vision method in structural health monitoring, template matching is typically used to trace deformation of structures. Olaszek [6] first used the template matching method to obtain the dynamic characteristics of bridges. Sładek [7] detected the in-plane displacement using template matching on a beam. Wu et al. [8] developed a vision system that uses digital image processing and computer vision technologies to monitor the 2D plane vibrations of a reduced scale frame mounted on a shake table. Busca et al. [9] proposed a vision-based displacement monitoring sensor system using three different template matching algorithms, namely, pattern matching, edge detection, and digital image correlation (DIC). Field testing was carried out to obtain the vertical displacement of a railway bridge by tracking high-contrast target panels fixed to the bridge.

Without template targets, computer vision methods can still be used to trace structural deformations. Poudel et al. [10] proposed an algorithm for determining the edge location with sub-pixel precision using information from neighboring pixels and used these edge locations to trace structural deformations. Wahbeh et al. [11] followed a similar approach to obtain direct measurements of displacement. Debella-Gilo et al. [12] proposed two different approaches to achieve the sub-pixel precision. In the first approach, a bi-cubic gray scale interpolation scheme prior to the actual displacement calculations was used. In the second approach, image pairs were correlated at the original image resolution followed with bi-cubic interpolation, parabola fitting, or Gaussian fitting to achieve sub-pixel precision. Chen et al. [13] proposed a digital photogrammetry method to measure the ambient vibration response at different locations using artificial targets. The mode shape ratio of stay cables with multiple camcorders was successfully identified. Feng et al. [14] proposed a vision sensor system for remote measurement of structural displacements based on an advanced subpixel-level template matching technique using Fourier transforms. 

The effectiveness of the computer vision method is affected by camera setup parameters and light intensity. Santos et al. [15] used the factorization method to get an initial estimate of the object shape and the camera’s parameters. Knowledge of the distances between the calibration targets was incorporated in a non-linear optimization process to achieve metric shape reconstruction and to optimize the estimate of the camera’s parameters. Schumacher and Shariati [16] developed a method that tracked the small changes in the intensity value of a monitored pixel to obtain the displacement. Park et al. [17] proposed a method based on machine vision technology to obtain the displacement of high-rise building structures using the partitioning approach. They performed verification experiments on a flexible steel column and verified that a tall structure can be divided into several relatively short parts so that the visibility limitation of optical devices could be overcome by measuring the relative displacement of each part. Chan et al. [18] proposed a charge-coupled device (CCD) camera-based method to measure the vertical displacement of bridges and compared the measured displacements with the results from optical fiber (FBG) sensors. It was concluded that both methods developed were superior to traditional methods, providing bridge managers with a simple, inexpensive, and practical method to measure bridge vertical displacements. Recently, Fioriti et al. [19] adopted the motion magnification technique for modal identification of an on-the-field full-scale large historic masonry structure by using videos taken from a common smartphone device. The processed videos unveil displacements/deformations hardly seen by naked eyes and allow a more effective frequency domain analysis.

Many existing image-based algorithms are based on template matching, and satisfactory results have been obtained with target templates as reviewed above. However, artificial targets sometimes are not easy to set up, and not all-natural targets are suitable to be high-contrast templates such as the cables of suspension bridges. Moreover, when the targets largely deform or break away from structures, they cannot be detected successfully. Furthermore, template targets need time to be prepared.

In this paper, a vision sensor system for monitoring the structural displacements based on advanced edge detection and subpixel technique is proposed. The performance of the proposed system is first verified through an MTS test. A field test is then carried out on a street sign, and the proposed vision system is used to provide the time history and natural frequencies of the street sign instantaneously. Finally, the use of such a system for damage detection in a steel beam is evaluated. 

## 2. Proposed Displacement Measurement Method 

Figure 1 shows the flowchart of the proposed displacement measurement method. A video of the vibration of a structure is first recorded using a high-speed camera. The captured video is then converted to images, which are further processed to grayscale. The edges in the images are calculated using the canny edge detector. A suitable edge at the measured point and its region of interest (ROI) are selected according to the position of the selected edge in the first image. The coordinates of the measured edge in every image are calculated using the proposed method at an integer pixel precision. Sometimes, the precision of the integer pixel is inadequate when the distance between the camera and the structure is large or the camera has a low resolution. Subpixels based on the Zernike moment developed in this paper is then used to obtain the displacements at the subpixel precision based on the integer pixel coordinates of the edge. 

An edge-detection method for monitoring displacement through images captured by a single camera is illustrated in Figure 2. Once the original images (Figure 2a) are obtained through the camera, they are converted to gray-scale images (Figure 2b). Edge detection using a Canny edge detector, which is defined on the integer pixel level, is then performed (Figure 2c). The white lines shown in Figure 2c indicate edges found while the background is set black through the greyscale thresholding process. Once the edges of a structure are found, a self-developed algorithm in MATLAB is executed to calculate the coordinates of the edges at measured points. Through this algorithm, displacements at any location of the target structure are measured. To reduce the runtime, the ROI is used to calculate displacements at certain locations of the structure. Most of the time, the displacements of a structure are small, so an image of the entire structure is not required. 

The proposed displacement measurement method is like a group of laser displacement sensors and LVDTs. The cameras are the reference points because they do not move, similar to fixtures of laser sensors or LVDTs. Every point on the edges of measured structures could be selected as there is a laser sensor at each point of the edges. It is also convenient to change measured points since they are captured simultaneously.

When the camera is not very close to the object or the camera does not have high resolution, the accuracy obtained is usually not adequate. There are two approaches to enhance it. One is to use a higher resolution camera. The other way is to use the sub-pixel technology to improve accuracy of the image analysis. Some researchers have incorporated the subpixel technique to the conventional template matching methodology, mostly using interpolation. In this paper, edge detection method is combined with the subpixel technique to achieve better accuracy. After the displacement at an integer-pixel level is obtained, the subpixel technique based on Zernike moment method could be used to obtain subpixel level displacements and achieve more accurate results. At the same time, to convert the displacement in pixel to the real physical distance, a relationship between the pixel and physical coordinates needs to be established.

## 3. Principles of Zernike Moment-Based Subpixel Edge Detection

Photo images are very sensitive to noises such as change in brightness and the vibration of the camera. Zernike moment is an integral operator that filters these effects and helps improve the displacement measurement accuracy. Zernike moment has the property of rotation invariance [20,21], which can be written as,
(1)$$Z_{n,m}^{\prime }=Z_{n,m}exp(-j\phi )$$
where $Z_{n,m}$ is the original *n* by m Zernike moment matrix; $Z_{n,m}^{\prime }$ is the transferred Zernike moment matrix; *j* is the imagery identifier; and ϕ is the rotation angle.

Figure 3 contains a model of the edge. Figure 3b is obtained when Figure 3a is rotated clockwise about the origin by *ϕ*. In the figure, *S* is the edge, *h* and *h + k* are the values of grayscale of two sides about *S*, *l* is the perpendicular length from the origin to *S* and *ϕ* is the angle between *l* and *x*-axis. In Equation (1) and Figure 2, *Z’_(n,m)* is the Zernike moment after the rotation. The exact coordinates of the edges will require *k*, *l*, and *ϕ*. Each Zernike moment element is defined in Equation (2).
(2)$$Z_{n,m}=\frac{n+1}{\pi }\iint_{x^{2}+y^{2}\leq 1}{\bar{V}}_{n,m}f\left(x,y\right)dxdy$$

$Z_{n,m}$ is the Zernike moment of f(x,y) at rank *n*. ${\bar{V}}_{n,m}$ is the conjugate function of $V_{n,m}$, the integral core function. The discrete form of Zernike moment can be written as in Equation (3).
(3)$$Z_{n,m}=\frac{n+1}{\pi {\left(N-1\right)}^{2}}\overset{N}{\underset{i=1}{\sum }}\overset{N}{\underset{j=1}{\sum }}f\left(x_{i},y_{j}\right){\bar{V}}_{n,m}\left(x_{i},y_{j}\right)$$
where *N* is the number of the integration points, *x_i_, y_j_* are the coordinates of the integration points, and *f(x_i_, y_j_)* is the grayscale of the pixel. 

$Z_{0,0}$, $Z_{1,1}$, and $Z_{2,0}$ are used to calculate the edges at subpixel level. Their integral core functions are ${\bar{V}}_{0,0}=1$, ${\bar{V}}_{1,1}=x-yj$, and ${\bar{V}}_{2,0}=2x^{2}+2y^{2}-1$, respectively (Table 1). 

The edge after rotation is symmetric about the x axis, so
(4)$$\iint_{x^{2}+y^{2}\leq 1}yf^{\prime }\left(x,y\right)dxdy=0$$
where $f^{\prime }\left(x,y\right)$ is the image function after rotation.
(5)$$Z_{1,1}^{\prime }=Z_{1,1}exp(-j\phi )$$
(6)$$Z_{1,1}^{\prime }=Re(Z_{1,1}^{\prime })+jIm(Z_{1,1}^{\prime })$$
where *Re* is the real part of $Z_{1,1}^{\prime }$, *Im* is its imaginary part.
(7)$$\phi ={tan}^{-1}(\frac{Im(Z_{1,1}^{\prime })}{Re(Z_{1,1}^{\prime })})$$
in which $Z_{1,1}^{\prime }$ and $Z_{2,0}^{\prime }$ are calculated using Equations (8) and (9),
(8)$$Z_{1,1}^{\prime }=\iint_{x^{2}+y^{2}\leq 1}f^{\prime }\left(x,y\right)\left(x-yj\right)dxdy=\iint_{x^{2}+y^{2}\leq 1}f^{\prime }\left(x,y\right)xdxdy=\frac{2k{\left(1-l^{2}\right)}^{\frac{3}{2}}}{3}$$
(9)$$Z_{2,0}^{\prime }=\iint_{x^{2}+y^{2}\leq 1}f^{\prime }\left(x,y\right)\left(2x^{2}+2y^{2}-1\right)dxdy=\frac{2kl{\left(1-l^{2}\right)}^{\frac{3}{2}}}{3}$$
*l* and *k* are calculated from Equation (10).
$$l=\frac{Z_{2,0}^{\prime }}{Z_{1,1}^{\prime }}=\frac{Z_{2,0}^{\prime }}{Z_{1,1}exp(-j\phi )}$$
(10)$$k=\frac{3Z_{1,1}^{\prime }}{2{\left(1-l^{2}\right)}^{\frac{3}{2}}}=\frac{3Z_{1,1}exp(-j\phi )}{2{\left(1-l^{2}\right)}^{\frac{3}{2}}}$$

When the values of *l, k, h* and ϕ are obtained, coordinates at subpixel level (asc and bsc) are calculated using Equation (11), where a and b are integer coordinates of the edge.
(11)$$\left[\begin{array}{c} a_{sc}\\ b_{sc} \end{array}\right]=\left[\begin{array}{c} a\\ b \end{array}\right]+l\left[\begin{array}{c} cos(\phi )\\ sin(\phi ) \end{array}\right]$$

An example test is conducted to verify the accuracy of the subpixel method using MATLAB (Figure 4). First, a circle with the center at (50, 75) and a diameter of 50 units is drawn using an equation. The color of the outside region is black, and the inside region is white. The line for the circle is the true edge between the black and white regions. The edges are calculated using both the integer-pixel edge detection and the subpixel edge detection, and they are compared with the real edge in red (Figure 4b). The improved accuracy of the subpixel method over the integer-pixel method is clearly seen when the detected edge is compared with the true edge. 

## 4. Lab Experiments and Results

### 4.1. Lab Experiments

Three tests were conducted in the lab to verify the proposed method. The tests were aimed at measuring the one-dimensional vibrations of the piston of an MTS machine. The MTS machine could provide up to a 3 Hz vibration and output the vibration data accurately. The displacement output by the MTS, whose error was less than 0.5% with respect to the given input command, was taken as the reference of the measurement methods. An accelerometer was also used to detect the vibrations, which were compared to the displacement data obtained through image processing. The acceleration data from the accelerometer will be integrated to obtain the velocity and displacement data using the detrended double integration method through a filter for frequencies higher than 60 Hz.

Figure 5 shows the experimental set-up in the lab. A XiaoYi commercial camera with 120 fps acquisition rate and 1920 × 1080 pixel resolution was placed 0.5 m from the MTS machine. A MEMS accelerometer was glued on the piston and connected with a laptop. The piston was mounted on the MTS machine, whose vibrations were computer controlled. Sinusoidal wave-type motions were adopted with controlled frequencies (2 Hz, 2.5 Hz and 3 Hz) and amplitudes (4 mm, 2 mm and 1.5 mm, respectively).

### 4.2. Results of Lab Experiments 

Figure 6a presents the results when a sinusoidal wave with amplitude of 4 mm with a frequency of 2 Hz was applied. Comparison of vibrations between the subpixel image processing, the integer pixel image processing, MTS, and the accelerometer measurements are shown in Figure 6b. Very good agreement between the subpixel image measurement and the displacement output by MTS is shown. Accuracy of the subpixel image processing method is better than that of the accelerometer method in monitoring displacement. For the integer image processing method, 56% of points per cycle are at the same value for the displacement shown. One possible reason is that the integer image processing method could only detect displacements at integer pixel values. In particular, when the piston moves slowly, more points at the same value of displacements will be obtained. 

The results of vibration measurements for the case with an amplitude of 2 mm and frequency of 2.5 Hz are shown in Figure 7a. Moreover, comparison of vibration measurements between the subpixel image processing method, MTS input, the integer pixel image processing method, and the accelerometer measurements are shown in Figure 7b. From Figure 7b, higher accuracy is achieved for the subpixel image processing method.

Figure 8a shows the results of vibrations measured by the subpixel image processing method, integer pixel image processing method, MEMS accelerometer, when a sinusoidal wave with amplitude of 1.5 mm and frequency of 3 Hz was applied. Comparisons of errors between the subpixel image processing and MTS, the integer pixel image processing and MTS, and the accelerometer measurements and the MTS are shown in Figure 8b.

According to Figure 6b, Figure 7b, and Figure 8b, the maximum error of subpixel method is less than 5%. Some measured displacements from the integer pixel method are larger than 10%. However, for MEMS accelerometer, the maximum error is typically larger than 20%. As shown in Figure 9, when the frequency is higher, larger errors are observed, because fewer points per cycle are obtained to describe the vibration. Higher resolution images could help but are also very expensive, so the subpixel image processing method is adopted instead. Moreover, the proposed subpixel image processing method does not need any targets on the object and can detect displacements at any location on the structure. In comparison to the template matching method, edge detection method is timesaving and avoids issues from events such as excessive deformation and failure of targets.

### 4.3. Field Test Monitoring the Vibration of a Street Sign

A field experiment was carried out using a street sign on the North Dakota State University (NDSU) campus to verify the subpixel image processing for edge detection. As shown in Figure 10, the street sign can be considered a cantilever column fixed on the ground. An accelerometer was attached on the street sign 40 cm above the ground to compare with the subpixel image processing method. The data acquisition rate of MEMS is 100 Hz. A moving averaging method is used to filter high frequency noises over 40 Hz in order to minimize the error. A camera was fixed on a tripod and set up 85 cm away from the street sign. A pulse was applied to the tip of the street sign to initiate the vibration. Figure 11 presents the vibration data from the image processing method. As shown in Figure 11, the vibration magnitude of the street sign was less than 2 mm.

The displacement time history and several natural frequencies were detected through the subpixel image processing method, although the amplitude of the street sign vibration is less than 2 mm. Figure 12 shows the modal frequencies obtained through the subpixel image processing. Through the proposed subpixel image processing, these frequencies are found at 7.85 Hz, 15.86 Hz, and 31.73 Hz. The proposed algorithm can be used at any location on the street sign since it has many edges and infinite points on each edge. Edge detection technique could avoid noises such as brightness changing, so the displacement in bad weather can also be detected. Subpixel methods could enhance the edge detection method and detect small displacements.

### 4.4. Identification of Damage through Image Analysis

The main purpose of the proposed method is to evaluate damages on structures based on changes in their natural frequencies. To do so, a laboratory experiment was conducted to track the changes in the natural frequencies of a steel beam. The natural frequency of the original steel beam with no damage was first measured using the proposed method. The beam has a dimension of 914.4 × 25.4 × 3.175 mm (36 × 1 × 1/8 in.). Then a rectangular incision was made on the steel beam and the natural frequency was measured again. The incision is made at 1 foot (0.305 m) away from the end with a depth of 0.5 in. (12.7 mm) and a width of 1.0 in. (25.4 mm). Finally, the location of damage was determined based on the differences between the natural frequencies of the original and damaged steel beam. 

To initiate vibration of the steel beam, an arbitrary manual tapping was applied. Figure 13 shows that the camera was fixed on a tripod and set up 1 foot (0.304 m) away from the steel beam. A MEMS accelerometer connected with a laptop was used to measure the displacement of the steel beam. Figure 14 shows the natural frequencies of the undamaged steel beam through the image processing method and MEMS measurements, while Figure 15 shows the natural frequencies of the damaged steel beam through the proposed method and the MEMS measurements. There is a gap between peaks obtained by MEMS and the subpixel image method, which is believed due to weight and large dimension of MEMS. The weight of MEMS reduces the natural frequency of the system, while large length of the senor averages the responses it covered and reduces the higher frequency data. The data in Figure 14 and Figure 15 was zoomed out to show the comparison of the first natural frequency, which will have the similar Power Spectrum Amplitude curve as shown in Figure 11 if the *x*-axis scale is set to 0–40 Hz. 

Through the two tests, the reduction of stiffness of the steel beam due to incision was successfully found using the proposed method. The natural frequency was reduced from 3.10 Hz to 2.88 Hz after the beam was damaged, even though the theoretical frequency calculated is estimated to be 3.24 Hz through $\frac{3.52}{2\pi }\sqrt{\frac{EI}{ml^{4}}}$, where *E* is the Young’s modulus of steel, *m* is the mass per unit length of the beam, *I* is the moment of inertia, and *l* is the length of the beam (the small difference between the measured natural frequency and the calculated natural frequency may be due to the clamped boundary condition applied). Thus, the proposed vision sensor can detect the reduction of stiffness of a structure. 

The location of the damage can also be identified if mode shapes of the structure could be found. Time domain decomposition (TDD) is a proven method used to extract mode shapes of a structure and identify structural damages, even though other methods such as neural networks shown in Mao et al. [23] could be also adopted for this purpose. Figure 16a shows the undamaged and damaged mode shapes. The two mode shapes are similar with small differences observed in the middle region. Normalized position is the ratio of the distance between the point on the bar and the left end (x) and the total length of the steel plate (L). The ratio of the displacement (d) and the maximum displacement (D) is defined as the normalized displacement. Figure 16b shows that there are some differences between the undamaged mode shape and damaged mode shapes. As shown in Figure 16b, the peak difference happened at a normalized location of 0.3 indicates the location of the damage. Through the analysis of mode shape difference, the damage was detected successfully. Since the mode shape and the damage detection method adopted here uses 11 points (10 segments), which makes the highest precision for the normalized damage location 0.1.

## 5. Conclusions and Future Work

In this study, a novel vision system was developed for noncontact displacement measurement of structures using a subpixel Zernike edge detection technique. Comprehensive experiments, including tests using an MTS machine, vibration tests on street sign and steel beams, were carried out to verify the accuracy of the proposed method. The following conclusions are reached:

1. In the MTS test, satisfactory agreements were observed between the displacement measured by the vision system and those by the MTS sensors. 

2. From the street sign test, vibration of the street sign after an excitation pulse was detected successfully. Using Fast Fourier Transform, several natural frequencies of the street sign were founded.

3. Subpixel-based Zernike matrix method is an innovative edge detection that could monitor structural displacements accurately at any location on the detected edges.

4. Through the analysis of mode shape obtained by the proposed image processing method, the damage location of the steel beam could be accurately detected. 

By tracking existing natural edges of structures, the vision sensor method developed in this paper provides flexibility to change measured locations on structures. The availability of such a remote sensor will facilitate cost-effective monitoring of civil engineering structures. 

The suggested method uses a camera with high resolution, which could be placed in a distant position to record the motion of target structures. The recorded videos could be then used to analyze the vibration and natural frequencies of the target bridge with sufficient accuracy. These scenarios are the real environment this method could be used. However, the flaws of this proposed method lie in the detection of damages in large structures. From mode shapes of a structure, one could find the structural conditions of the target (whether it has potential problems). However, to determine the damage location and its severity in terms of sizes and shapes, the current method needs further improvement.

## Figures and Tables

**Figure 1 sensors-20-04941-f001:**
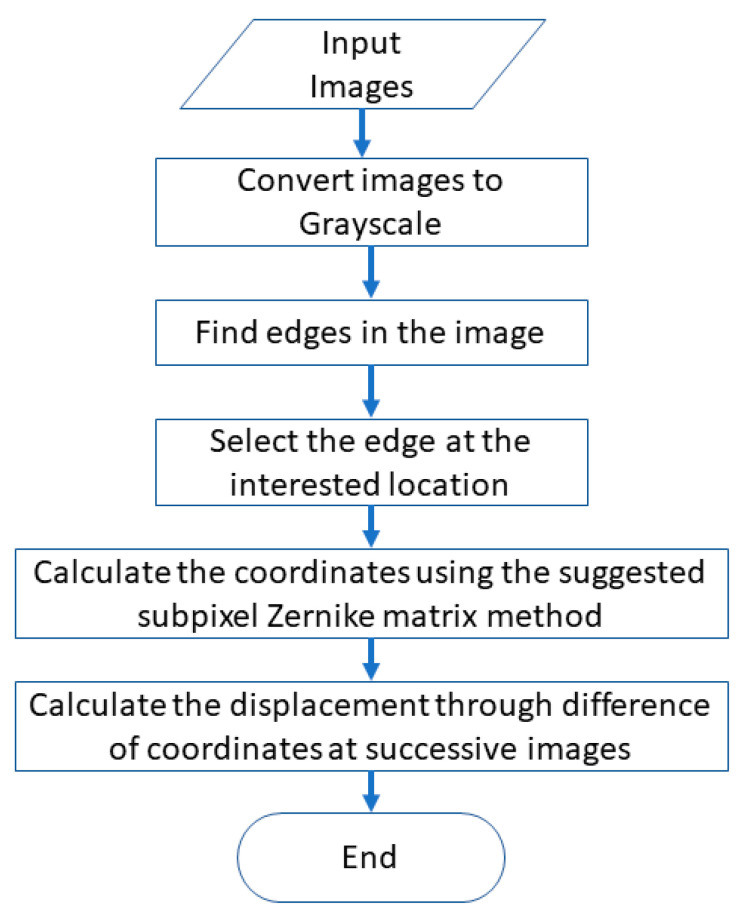
Flowchart of the proposed image-based vision sensor method.

**Figure 2 sensors-20-04941-f002:**
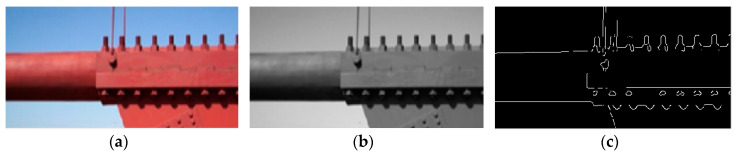
Processed image: (**a**) Original image; (**b**) Grayscale image; (**c**) Edges in the image.

**Figure 3 sensors-20-04941-f003:**
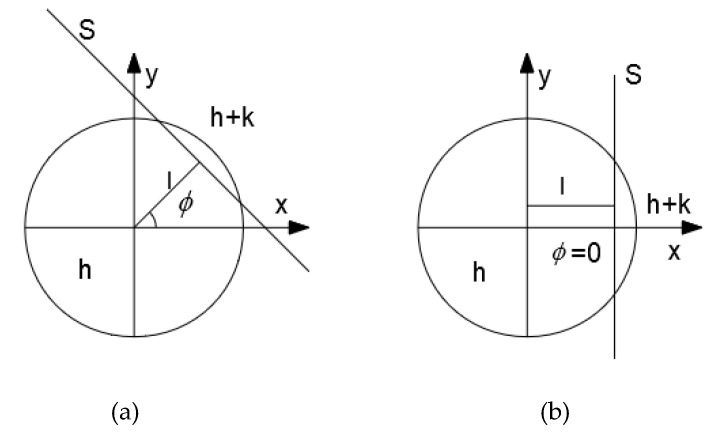
Model of edge: (**a**) Original edge and (**b**) Edge after rotation.

**Figure 4 sensors-20-04941-f004:**
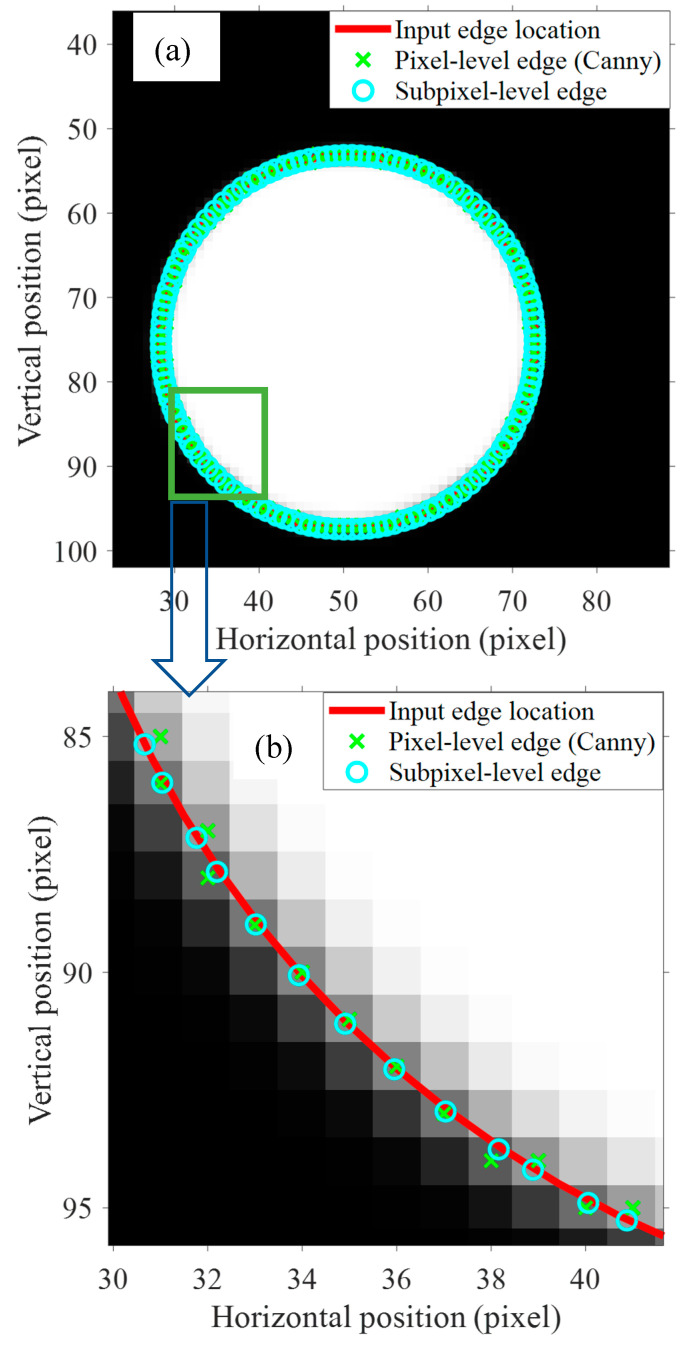
Performance of subpixel method and integer-pixel method. (**a**) The overall image, (**b**) Zoom-in detail of (**a**).

**Figure 5 sensors-20-04941-f005:**
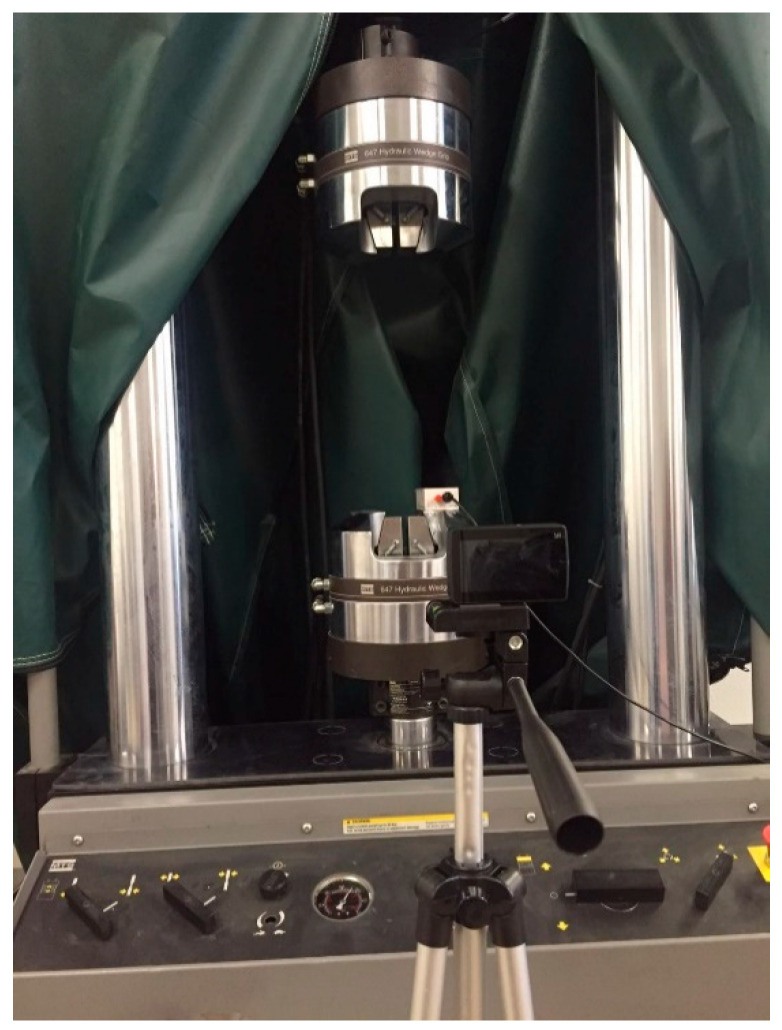
Set-up for lab experiment.

**Figure 6 sensors-20-04941-f006:**
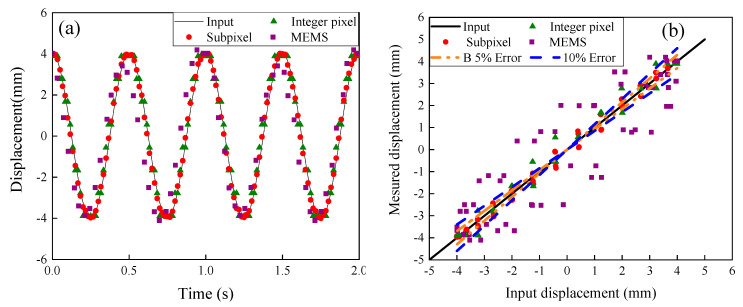
(**a**) Comparison of captured displacements with the input vibration with a frequency of 2 Hz and amplitude of 4 mm, and (**b**) Error analysis.

**Figure 7 sensors-20-04941-f007:**
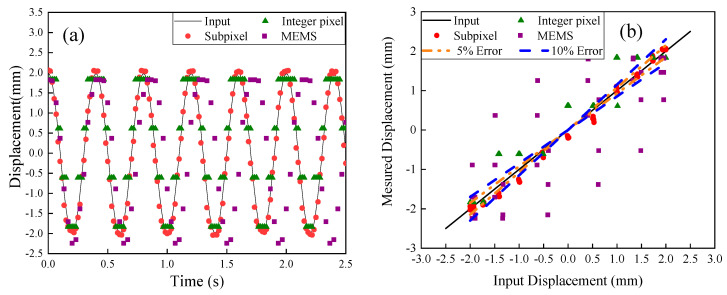
(**a**) Comparison of captured displacement with the input vibration at a frequency of 2.5 Hz and amplitude of 2 mm, and (**b**) Error analysis.

**Figure 8 sensors-20-04941-f008:**
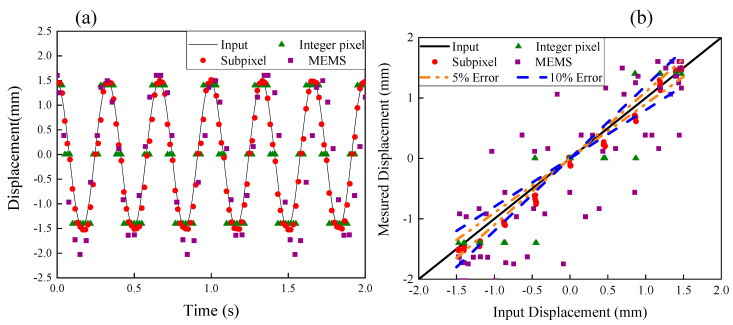
(**a**) Comparison of captured displacement data with input sinusoidal function with frequency of 3 Hz and amplitude of 1.5 mm and (**b**) Error analysis.

**Figure 9 sensors-20-04941-f009:**
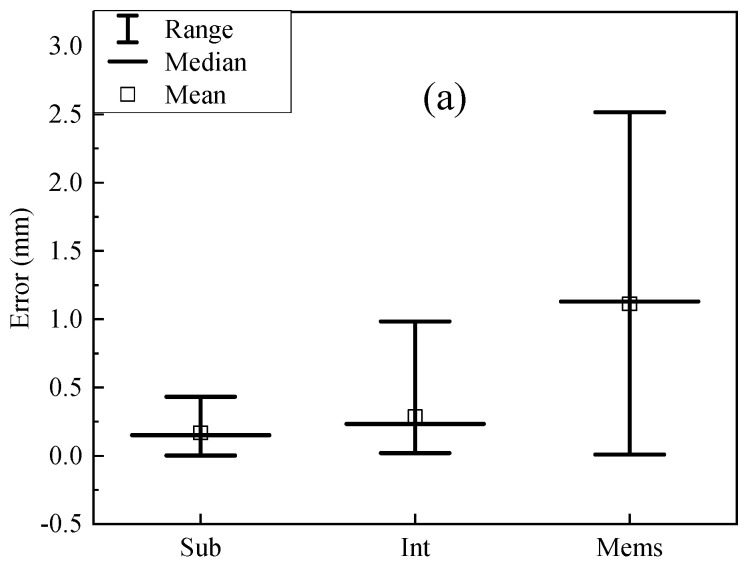
Error analysis of experiments: (**a**) 4 mm at 2 Hz; (**b**) 2 mm at 2.5 Hz; (**c**) 1.5 mm at 3 Hz (Sub—subpixel method, Int—integer pixel method, Mems—MEMS accelerometer method).

**Figure 10 sensors-20-04941-f010:**
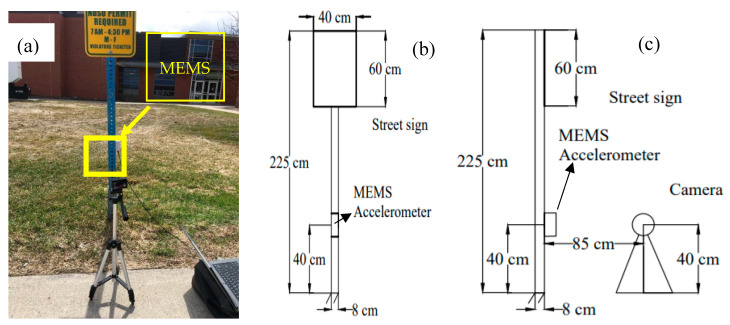
Setup of street sign experiment: (**a**) Field photo, (**b**) Front view with dimensions, (**c**) Side view with dimensions.

**Figure 11 sensors-20-04941-f011:**
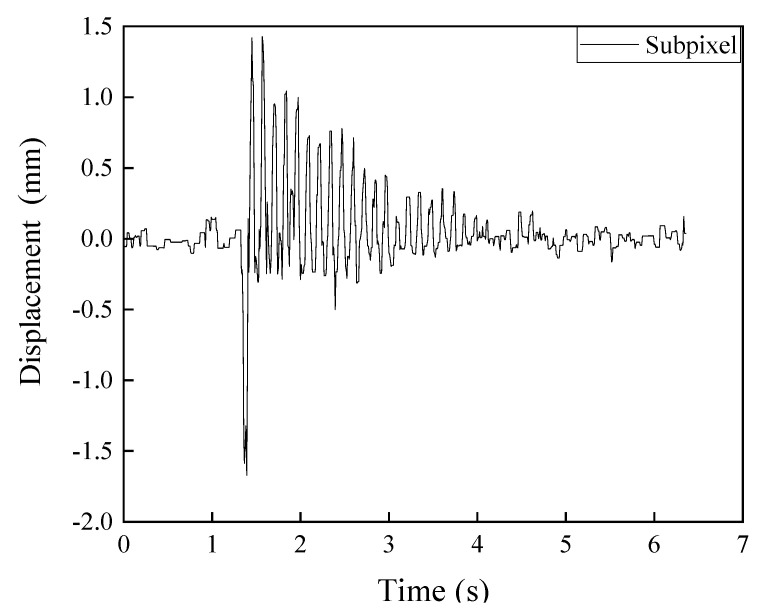
Vibration of street sign using subpixel image processing.

**Figure 12 sensors-20-04941-f012:**
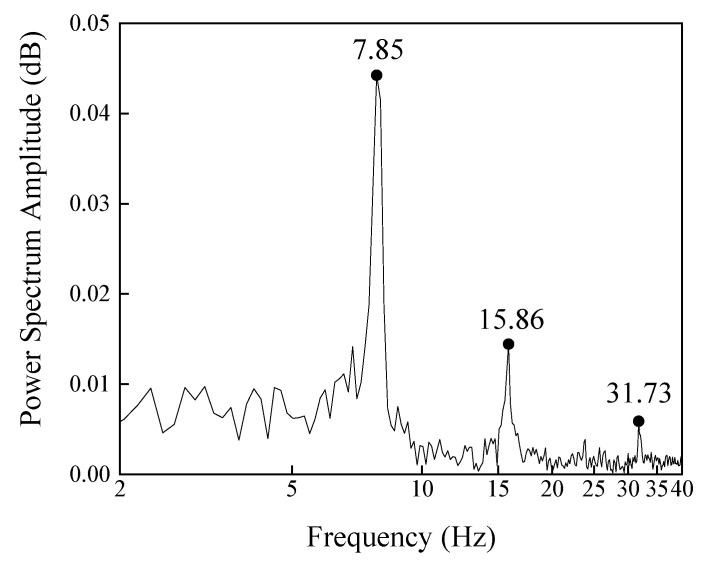
Modal frequencies of street sign at measured location.

**Figure 13 sensors-20-04941-f013:**
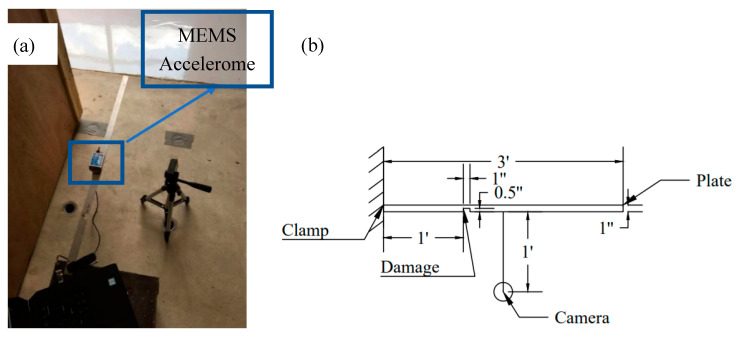
(**a**) Setup of the steel beam vibration test, (**b**) Schematic diagram of the experiment setup (1’ = 12’’ = 0.3048 m).

**Figure 14 sensors-20-04941-f014:**
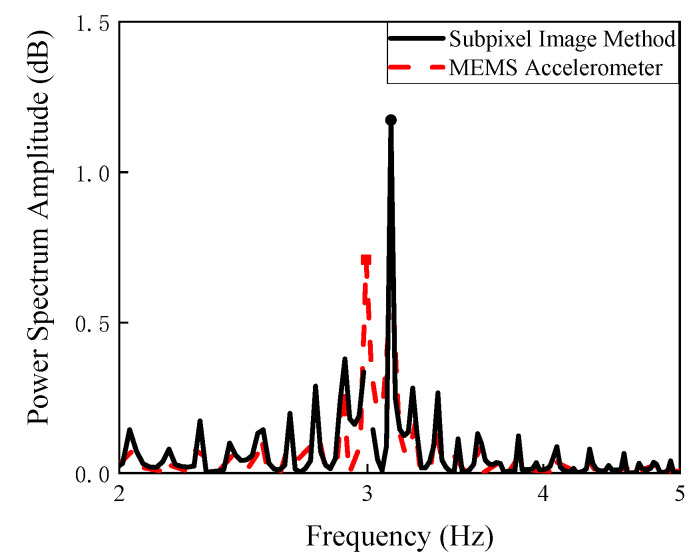
Natural frequency of the undamaged steel beam through the MEMS accelerometer and subpixel image method.

**Figure 15 sensors-20-04941-f015:**
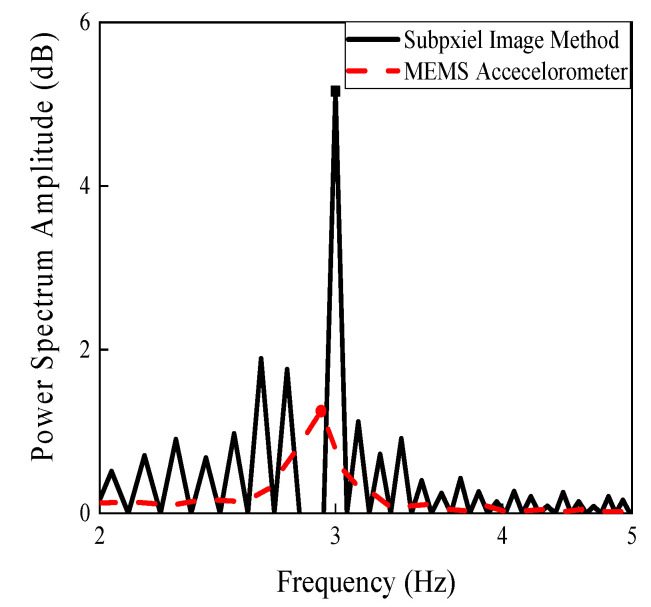
Natural frequency of the damaged steel beam through the MEMS accelerometer and subpixel image method.

**Figure 16 sensors-20-04941-f016:**
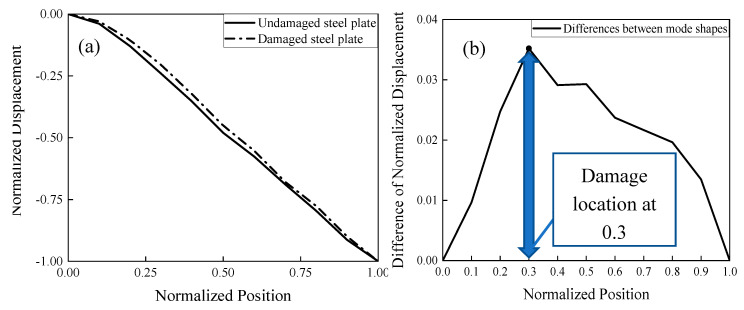
Mode shape analysis. (**a**) Mode shapes of the undamaged and the damaged steel beam. (**b**) Differences in the normalized displacement between the undamaged mode shape and damaged mode shape.

**Table 1 sensors-20-04941-t001:** Zernike orthogonal complex polynomials (${\bar{V}}_{n,m}$) [22].

*m/n*	0	1
0	1	/
1	/	x−yj
2	$2x^{2}+2y^{2}-1$	/
3	/	$\left(3x^{3}+3xy^{2}-2x\right)+\left(3y^{3}+3x^{y}-2y\right)j$
4	$6x^{4}+6y^{4}+12x^{2}y^{2}-6x^{2}-6y^{2}+1$	/

## Data Availability

Some or all data, models, or code generated or used during the study are available from the corresponding author by request, which include: (1) The edge detection code; (2) The experimental images.

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
