# Peer review of "An Advanced Edge-Detection Method for Noncontact Structural Displacement Monitoring"

_sensors, 2020, doi:10.3390/s20174941_

Round 1

Reviewer 1 Report

  • In the introduction, explain the difference and innovation of the proposed method from other methods.
  • The explanation of the proposed displacement measurement method is too simple. It is recommended to combine references or diagrams to explain in more detail the innovation and processing flow of the proposed method.
  • In the experimental part, the corresponding references should be cited for comparison with other methods, for example, the integer pixel image processing, MTS, and the accelerometer measurements. Whether there is a public experimental environment as a benchmark, and compare the various methods based on the benchmark. If so, it is recommended to add this part of the experiment. The paper must compared with state-of-the-arts edge detection methods.
  • It is recommended to adjust the format of formulas in the paper. Please double check the typesetting, especially formulas.

Reviewer 2 Report

In this paper, a vision sensor system for displacement monitoring of structure based on edge detection and subpixel technique is proposed, and two tests were carried out to verify the method. In my opinion, the manuscript should be improved to get more results to meet real applications.

Line 204 to 206, there are reviewer's comments?

The field test need to be improved to meet the real requirement, and more results should be given under different situation, e.g., both large and small target, long and short distances between the sensor and the target to test the precision...

Reviewer 3 Report

In this paper, the author presented a vision sensor system for monitoring the structural displacements based on advanced edge detection and subpixel technique. Some problems need to be addressed before the reviewer thinks the manuscript can be published.

  1. Workflow of vision sensor system is recommended to be included in the beginning of the paper.
  2. In Figure 3(b), why there are grey pixels between black and white regions? Is the figure artificially produced?
  3. ‘Subpixel based Zernike matrix method is an innovative edge detection that could monitor structural displacements accurately at any location. This conclusion may be applicable to most of image processing methods derived based on pixel-based tracking.
  4. The quality of formula and figures should be improved for much clearly, such as Figure 12(a).
  5. If the distance between object and camera increase, to what extent will the accuracy of subpixel technique be? It is advised to carried out researches and comparisons to see whether the subpixel technique has the advantages over integer pixel technique.
  6. Related references are suggested to be included in the revised manuscript. Fatigue reliability assessment of a long-span cable-stayed bridge based on one-year monitoring strain data, Automated modal identification using principal component and cluster analysis: Application to a long‐span cable‐stayed bridge; Toward data anomaly detection for automated structural health monitoring: Exploiting generative adversarial nets and autoencoders

Reviewer 4 Report

The paper presents a study on a novel method to improve displacement monitoring of structures through video footage. In particular, the innovative method is focused on improving the algorithm for digital video processing in order to obtain a more accurate measurement of structure vibrations with the aim at providing accurate modal parameters, which are very important for the so-called structural health monitoring. The proposed algorithm is described in depth and three different experimental applications to real vibrating objects in laboratory and in outdoor environment are illustrated as case studies in order to demonstrate the method performance in comparison with more conventional measurement methods. The overall manuscript is sufficiently well written and organized. English is good. However, some improvements of the paper should be considered by the authors. Some suggestions are as follows:    

Lines 58-71: here authors review computer vision methods for monitoring structural vibrations without template targets. They should add a recent remarkable application derived from the motion magnification technique for modal identification of an on-the-field full-scale large historic masonry structure by using videos taken from common smartphone device. As large structures are usually characterized by fundamental frequencies lower than 5 Hz, encouraging results could be obtained also with 30-60 fps devices. The authors can make reference to the following paper:

  • Fioriti, V.; Roselli, I.; Tatì, A.; Romano, R.; De Canio, G. Motion Magnification Analysis for Structural Monitoring of Ancient Constructions. Measurement 2018, 129, pp. 375-380, ISSN: 0263-2241, DOI: 10.1016/j.measurement.2018.07.055

Lines 204-206: these lines look like suggestions from a reviewer. Please, delete them and possibly follow the suggestions…

Lines 207-210: it is clear that the authors compared several measurement methods: camera measurement with proposed sub-pixel method, camera measurement with integer pixel method, MTS input, and accelerometer with detrended double integration method. What is not clear, and not obvious at all, is the accuracy of the MTS input, which is considered by the authors the reference real motion. In fact, it is not obvious that the machine reproduces the input motion with a better accuracy than the considered measurement methods. From a methodological point of view it is not correct and the authors should compare the accuracy of the new method only to data obtained with measurement methods having known accuracy. To remedy to this methodological issue, please, provide information on the MTS input reproduction accuracy or on the other methods accuracy for a reasonable comparison with the new method performance.

Section 4.3: similarly to the previous comment, here the new method performance is compared to another measurement method (MEM accelerometer with double integration) whose accuracy is not provided. Please provide accuracy of used MEM accelerometer.

Lines 271-273: why didn’t the authors calculate the modal frequencies of the street sign also through the accelerometer signal? It would be very interesting to compare to the found frequencies.

Section 4.4: in figures 13 and 14 the frequency change in the steel beam after damage is assessed through MEMS accelerometer and the camera video data processed by the proposed subpixel method. Surprisingly, the subpixel image method overestimated the frequency in both cases. Do authors have an explanation for this? in fact, the Power Spectrum Amplitude by subpixel image method looks quite inaccurate. Maybe the authors can try some techniques to improve the extraction of the frequencies from the subpixel image method displacement data. For example, they can try with a simple moving average filter in order to produce a smoother and less noisy Power Spectrum Amplitude curve. Anyway, the Power Spectrum Amplitude curve shown in Figure 11 looks much better, can you explain why?

Lines 307-310: what are the frequencies found through the proposed subpixel image method? Can you estimate the percentage error with respect to the frequencies by MEMS accelerometer?

Figure 15(b) has something wrong. The blue arrow indicating the damage location must be fixed. Please, provide numerical value of the estimated normalized position of damage and error with respect to real damage position.

Section 5. conclusions: point 3 at lines 337-338 is quite vague. Please, clarify the meaning of “…could monitor structural displacements accurately at any location.”

Round 2

Reviewer 1 Report

From my point of view, I suggest some minor modifications before publication:

1)    Fig 1 is all words, it should be revised.

2)    It is recommended to modify the picture layout of the experimental part, for example, put (a) and (b) of Fig 6 in one row.

Reviewer 2 Report

I  do not think the proposed method has potential application in bridge etc. the real enviroment could be totally different.

Reviewer 3 Report

Two extra problems need to be addressed before the manuscript can be accepted.

  1. The format and quality of workflow chart of vision sensor system can be improved more aesthetically.
  2. A gap between the peaks identified by subpixel image method and MEMS accelerometer can be clearly observed, as shown in Fig. 14. Necessary descriptions and explanations are required.

Author Response

Please send the attachment.

Reviewer 4 Report

Comment # 3, line 235: for addressing the authors should add explicitly here the following sentence:
"The displacement output by MTS, whose error is less than 0.5% with respect to the given input command, is taken as reference of the measurement methods."

Comments # 4 and # 5. Section 4.3: From the authors replies this reviewer understands that the MEMS accelerometer used in the vibration measurements of the street sign was unable to provide any valid result, either in terms of displacement time-history or of modal frequencies. If so, please, state it explicitly in the text and provide a possible explanation. However, the authors should provide the used MEMS accelerometer accuracy or technical specifications, as well as the results of MEMS measurements. Moreover, the authors should state also explicitly the sampling rate of MEMS accelerometers and their data processing method in the text. For example, at line 287 after "method." the authors should add as follows:
"The rate of the data acquisition of MEMS is 100Hz. A moving averaging method is used to filter high frequency noises over 40 Hz in order to minimize the error."

In comment # 6 the authors state that the “proposed image method is more accurate from the MTS test results in section 4.2.”. Then this reviewer does not see any valid comparison between the proposed method and any consolidated accurate method for the estimate of modal frequencies… how could the performance of the proposed method be assessed by comparing its results with a less accurate method? This point is highly controversial. To remedy to this point, the authors should, at least, calculate the undamaged frequency of the steel beam according to the theory (it is a very simple exercise, once steel beam geometry and material properties are known) and compare the frequency found by the proposed method with the calculated theoretical undamaged frequency.
Comment # 7: the authors should mention explicitly the values of frequencies estimated by using the MEMS and explain clearly the meaning of a comparison with them.
